# Effect of a Targeted Ambulance Treatment Quality Improvement Programme on Outcomes from Out-of-Hospital Cardiac Arrest: A Metropolitan Citywide Intervention Study

**DOI:** 10.3390/jcm12010163

**Published:** 2022-12-25

**Authors:** Xuejie Dong, Liang Wang, Hanbing Xu, Yingfang Ye, Zhenxiang Zhou, Lin Zhang

**Affiliations:** 1School of Public Health, Shanghai Jiao Tong University, Shanghai 200025, China; 2Department of Global Health, Peking University School of Public Health, Beijing 100191, China; 3Suzhou Emergency Center, Suzhou 215002, China; 4School of Nursing, Shanghai Jiao Tong University, Shanghai 200025, China

**Keywords:** out-of-hospital cardiac arrest, quality improvement, advanced life support, ROSC, ambulance crew

## Abstract

The performance of ambulance crew affects the quality of pre-hospital treatment, which is vital to the survival for out-of-hospital cardiac arrest (OHCA) patients, yet remains suboptimal in China. In this retrospective analysis study, we aimed to examine the effect of a citywide quality improvement programme on provision of prehospital advanced life support (ALS) by emergency medical service (EMS) system. EMS-treated adult OHCA patients after the implementation of the programme (1 January 2021 to 30 June 2022) were compared with historical controls (1 June 2019 to 31 August 2020) in Suzhou. Multivariable logistic regression analysis and propensity score matching procedures were applied to compare the outcomes between two periods for total OHCA cases and subgroup of cases treated by fixed or non-fixed ambulance crews. A total of 1465 patients (pre-period/post-period: 610/855) were included. In the 1:1 matched analysis of 591 cases for each period, significant improvement (*p* < 0.05) was observed for the proportion of intravenous (IV) access (23.4% vs. 68.2%), advanced airway management (49.2% vs. 57.0%), and return of spontaneous circulation (ROSC) at handover (5.4% vs. 9.0%). The fixed ambulance crews performed better than non-fixed group in IV access and advanced airway management for both periods. There were significant increases in IV access (AOR 12.66, 95%CI 9.02–18.10, *p* < 0.001), advanced airway management (AOR 1.67, 95% CI 1.30–2.16, *p* < 0.001) and ROSC at handover (AOR 2.37, 95%CI 1.38–4.23, *p* = 0.002) after intervention in unfixed group, while no significant improvement was observed in fixed group except for IV access (AOR 7.65, 95%CI 9.02–18.10, *p* < 0.001). In conclusion, the quality improvement program was positively associated with the provision of prehospital ALS interventions and prehospital ROSC following OHCA. The fixed ambulance crews performed better in critical care provision and prehospital outcome, yet increased protocol adherence and targeted training could fill the underperformance of non-fixed crews efficaciously.

## 1. Introduction

Out-of-hospital cardiac arrest (OHCA) is a major public health problem globally with high mortality, high morbidity, and large variation in survival between communities [1]. In China, the incidence of cardiac arrest was estimated to be 550,000 cases per year and the survival was only 1–2% [2,3]. The survival outcome of OHCA is largely determined by the timelines and performance of pre-hospital interventions, known as the “chain of survival”, and could be affected by patient factors, event factors, and system factors of the emergency medical service (EMS) system [4]. The EMS is not only the first healthcare encounter providing advanced life support (ALS) to OHCA patients, but also works its way through monitoring of all the prehospital links in the chain of survival [5,6]. Hence, enhancing EMS performance and quality is fundamental for improving OHCA outcomes.

Various systemic attempts have been made to optimize the timelines and quality of EMS performance [7,8]. Studies have shown improved patient outcomes with higher adherence to guideline components within EMS systems, including EMS performed high-quality cardiopulmonary resuscitation (CPR), team-based training, structured resuscitation choreography, training, feedback, and profiling [9,10,11,12,13]. Quality improvement programmes focusing on above components have been shown to be associated with improved EMS performance and increased patient survival [10,14].

The performance of EMS personnel is a modifiable factor that may also influence patient outcome. Studies have found that ambulance crew staffing models and numbers [15,16], staffing patterns [17], level of training or individual experience [18,19] all affect the ambulance crew performance and outcome. The EMS in China has adopted a supplementary physician or physician-paramedic model for ambulance advanced care: the prehospital medical crew of the urban EMS typically consists of one physician, one nurse, one stretcher-bearer, and one driver [20,21]. The crew configuration and the number of personnel on the ambulance are always the same. It was hypothesized that a fixed ambulance crew (i.e., the crew members remain the same) would perform better than the unfixed one (i.e., the crew members would be changed) with more tacit understanding of each other and better team coordination. However, there is relatively little evidence to address the differences in the quality of care and patient outcomes performed by fixed or unfixed ambulance crew.

In China, the survival rate of OHCA remains poor for years, and new interventions are needed for further optimization. Suzhou EMS first employed the dispatcher-assisted CPR (DA-CPR) protocols in China from 2010, and equipped itself with improved information technology [22]. In 2020, based on evidence that higher adherence to guidelines, targeted training, monitoring and feedback, and coordinated ambulance crews improve OHCA outcomes, Suzhou EMS implemented a 4-month (from 1 September 2020 to 30 December 2020) citywide quality improvement programme, including a standardization of ambulance treatment protocol, targeted training of ambulance crew combined with quality monitoring and feedback throughout the whole process.

The purpose of this study was to examine the impact of this quality improvement programme on provision of prehospital ALS for OHCA patients and prehospital return of spontaneous circulation (ROSC). Moreover, we tested the hypothesis that fixed ambulance crew would be correlated with better performance and patient outcome.

## 2. Materials and Methods

### 2.1. Study Design

This was a retrospective analysis of prospectively collected EMS registry data of Suzhou EMS from 1 July 2019 to 30 June 2022, to investigate the effects of quality improvement interventions on ALS treatment and outcome of OHCA patients. Adult OHCA patients (aged ≥ 18 years) receiving EMS treatment after the implementation of the quality improvement programme between 1 January 2021 and 30 June 2022 were compared with historical controls between 1 July 2019 and 31 August 2020. Cases during the COVID-19 epidemic periods (1 February 2020 to 30 April 2020, and 1 February 2022 to 30 April 2022) were excluded.

### 2.2. Setting

Suzhou EMS center services a population of about 6.7 million people across 2996 square kilometers in 6 urban districts with an annual emergency call volume of ≈412,000 and annual ambulance dispatch volume of ≈144,000. The EMS center consists of 157 physicians, 176 nurses, 83 stretcher-bearers, and 171 drivers responding from 45 stations (1 directly affiliated station and 44 network stations), and works its way through dispatching, prehospital transport, and treatment of patients. The Medical Priority Dispatch System (MPDS) protocol is used by Suzhou EMS since 2011. For suspected cardiac arrests, DA-CPR is delivered and the nearest available ambulance unit is dispatched to perform ALS care.

Each ambulance unit consists of one physician, one nurse, one stretcher-bearer, and one driver. According to the way physician and nurse are partnered, the ambulance units were classified into two types: fixed and unfixed. In a fixed unit, one physician would always work with the same nurse, and they may cooperate with specific stretcher-bearer/driver or not. In an unfixed unit, there is no specific partnership between physician and nurse, one physician would work with different nurses on every duty, and they may cooperate with specific stretcher-bearer/driver or not. It should be mentioned that the physician in a fixed unit is a dedicated prehospital personnel who only works on ambulance, while in an unfixed unit, the physician could be dedicated prehospital personnel or rotated from in-hospital emergency department or other in-hospital departments. During the study period, there were 11 fixed ambulance crew stations and 34 unfixed stations in Suzhou EMS.

### 2.3. Intervention

In 2020, Suzhou EMS implemented a 4-month (from 1 September 2020 to 31 December 2020) citywide quality improvement programme consisting of the following: (1) standardized ambulance treatment protocol adopted, (2) ambulance crew targeted training, (3) quality monitoring, feedback, and post-event debriefing.

#### 2.3.1. Standardized Ambulance Treatment Protocol Adopted

On 1 September 2020, Suzhou EMS launched a prehospital standardized ambulance treatment protocol for OHCA patients, as shown in Figure 1, which was the first ALS treatment protocol for ambulance crews. Previously, although the ambulance crews performed according to the treatment guidelines, there was no unified ALS team choreography, and each team mainly operated according to their own experience. It was usual that the physician on the ambulance dictated the treatment measures, which could lead to a large variation in ALS treatment quality. The standardized treatment protocol was developed in line with the updated ALS guidelines (further modified based on the 2020 American Heart Association (AHA) guideline in October 2020), combined with the best practices of well-performing local EMS stations. The standardized treatment protocol highlighted the flow of ALS interventions, the specific role for each member of ambulance crews, and the timing for hospital transfer or termination of resuscitation. Consistent with the indeterminate evidence regarding duration of resuscitative efforts, no time limit was given for field attempts, but it was suggested that all appropriate resuscitation efforts be made and that efforts should continue until (1) the patient be transported if good prognostic signs were present after field efforts; (2) the declaration of death; or (3) strong demand for transfer from the patient’s families.

#### 2.3.2. Ambulance Crew Targeted Training

All personnel of EMS stations were instructed in the standardized protocol using lectures and simulated practices over a 4-month period. They underwent live didactic training on the updated AHA guidelines, the standardized protocol, and its key indicators measures. Experiences from the best-performing EMS stations were shared. An adding lecture was held after the 2020 AHA ALS guideline update released in October 2020. In the practice sessions, all ambulance crews performed hands-on simulated resuscitations using mannequins with feedback function, supervised by the medical director and medical division officers.

#### 2.3.3. Quality Monitoring, Feedback, and Post-Event Debriefing

The EMS medical director (L. Wang) performed a post-incident review of all cardiac arrests. A monthly feedback report was sent to each EMS stations and every ambulance crew involved in the resuscitation at the end of each month. Direct feedback was provided to the ambulance crew in person if any corrective action was needed.

### 2.4. Data Source

This study used data from the MPDS system registry and pre-hospital electronic medical record system of Suzhou EMS. The MPDS system registry is generated automatically along with the dispatching process and recorded the patient’s age, gender, location, chief complaint, dispatcher suspected disease, and the duration of each phase of the dispatch instructions. The pre-hospital electronic medical record is filled by ambulance physicians at the field using a tablet computer, and completed after transporting the patients to in-hospital emergency department, which contains patient’s symptoms, field diagnosis, disease history, treatment measures, pre-hospital outcome, time intervals, EMS station and ambulance personnel.

### 2.5. Study Population

All EMS-treated patients with OHCA between 1 July 2019 and 30 June 2022 were included. Patients who were confirmed dead at scene, whose resuscitation was not initiated or refused, who were only transferred from one hospital to another, pediatric patients younger than 18 years of age, and those cases in which time variables were missing or treated by stations built after the quality improvement programme were excluded. There were two COVID-19 epidemic periods in Suzhou: 1 February 2020 to 30 April 2020 and 1 February 2022 to 30 April 2022, respectively. The ambulance treatment protocol during epidemic was different from that of epidemic-free period, thus OHCA cases occurring in these two periods were excluded (Figure 2).

### 2.6. Data Collection and Outcome Measures

The primary outcome was the proportion of ROSC at handover at hospital. The secondary outcomes were prehospital ALS interventions by ambulance crews, including defibrillation, intravenous (IV) access, IV epinephrine administration, and advanced airway management. Outcomes were obtained from the pre-hospital electronic medical record system.

The following variables of interests were collected: patient demographics (age, gender), event characteristics (location of arrest, date and time of arrest, cardiovascular disease (CVD) history, etiology, onset-to-call time [from symptom onset to call receiving time]), EMS system factors (response interval [from call receiving time to scene arrival time of the ambulance crew], scene interval [from scene arrival time of the ambulance crew to departure time from the scene], transport interval [from the departure time from the scene to emergency department (ED) arrival time], prehospital interval [from call receiving time to ED arrival time], ambulance crew type), prehospital ALS interventions (EMS defibrillation, IV access, IV epinephrine administration, advanced airway management), intervention period (pre- and post-intervention period), prehospital ROSC (ROSC at handover).

### 2.7. Statistical Analysis

The patient demographics, event characteristic, EMS system factors, prehospital ALS interventions, and prehospital ROSC were compared between pre-intervention and post-intervention period. Normal distribution was confirmed using the Kolmogorov–Smirnov test. Continuous variables with normal distribution were shown as mean (standard deviation, SD), and compared by student *t* test. Continuous variables with non-normal distribution were shown as median (interquartile range, IQR), and compared using Wilcoxon test. Categorical variables were presented as the number (percentage), and compared using chi square test.

To compare the proportions of prehospital ALS interventions and ROSC at handover at hospital between two groups, both propensity score-matching analysis and logistic regression analysis were conducted to adjust for selection bias. In propensity score-matching analysis, we calculate estimated propensity score by fitting a logistic regression model that included age, gender, location, onset on weekend, call time of the day, CVD history, etiology, onset-to-call-time, call-to-arrival time, on-scene time, scene-to-hospital time, call-to-hospital time, ambulance crew type. We performed 1:1 matching without replacement for each patient, using a nearest-neighbor matching algorithm with a caliper width of 0.02. Matched patients were compared to assess balance in covariates (i.e., standardized differences for each covariate were <10%). For total OHCA cases and subgroup of fixed and non-fixed ambulance crew, a multivariable logistic regression model adjusted for age, gender, location, onset on weekend, call time of the day, CVD history, etiology, onset-to-call time, call-to-arrive time was used to compare the proportions of prehospital ALS interventions between the two groups to calculate the adjusted odds ratio (AOR) and 95% confidence interval (95% CI). Another multivariable logistic regression model adjusted for age, gender, location, onset on weekend, call time of the day, CVD history, etiology, onset-to-call time, call-to-arrive time, on-scene time, scene-to-hospital time was used to compare the proportion of ROSC at handover at hospital.

A two-sided significance level of less than 0.05 was considered statistically significant. All statistical analyses were performed using R software version 4.0.4.

## 3. Results

### 3.1. Patient Characteristics

A total of 2599 OHCA patients were treated by Suzhou EMS during the study period. The EMS attended and treated OHCA incidence rates of urban districts in Suzhou were 11.8, 13.0, 13.2, 13.2 per 100,000 population per year from 2019 to 2022. After exclusion of patients during the intervention-period and COVID-19 epidemic periods, 1465 patients were included as total study population. Of these, 610 (41.6%) and 855 (58.4%) occurred during the pre- and post-intervention periods, respectively. All included patients were witnessed and called for help by bystanders, attended and treated by EMS, all received CPR by EMS on-scene and during transport.

The patient demographics and event factors were similar between two periods. The post-period group showed longer scene interval [5.6 (3.0, 9.2) > 5.2 (2.6, 8.5) min, *p* = 0.033], longer transport interval [8.8 (5.3, 14.2) > 7.1 (4.4, 10.3) min, *p* < 0.001], longer prehospital interval [28.7 (22.0, 36.2) > 25.7 (20.8, 33.4) min, *p* < 0.001], and higher proportion of non-fixed ambulance crew (74.5% vs. 69.3%, *p* = 0.033) than the pre-period (Table 1).

### 3.2. Main Analysis

After propensity score matching, 591 patients of both groups were matched, the post-matching standardized mean differences were <10% for all matching covariates. In matched cases, the post-intervention group showed higher proportion of IV access (68.2% vs. 23.4%, *p* < 0.001) and advanced airway management (57.0% vs. 49.2%, *p* = 0.009), lower proportion of EMS defibrillation (3.0% vs. 5.8%, *p* = 0.032). The ROSC rate at handover at hospital was significantly improved after the intervention (9.0% vs. 5.4%, *p* = 0.024) (Table 1).

In the multivariable logistic regression model of total cases, the intervention was associated with higher proportion of IV access (AOR: 7.23; 95%CI: 5.69–9.24, *p* < 0.001), advanced airway management (AOR: 1.41; 95%CI: 1.14–1.75, *p* = 0.002), and ROSC at handover at hospital (AOR: 1.81, 95%CI: 1.18–2.84, *p* = 0.008) (Figure 3).

### 3.3. Subgroup Analysis by Types of Ambulance Crew

Table 2 and Figure 3 showed the results of subgroup analysis by types of ambulance crew. For prehospital ALS interventions, the proportions of IV access and advanced airway management were significantly higher in fixed group than non-fixed group both in pre-period and post-period. The multivariable logistic regression results suggested a significant increase in the proportion of IV access in fixed group (AOR: 7.65; 95% CI: 4.70–12.79, *p* < 0.001), proportions of IV access (AOR: 12.66; 95%CI: 9.02–18.10, *p* < 0.001) and advanced airway management (AOR: 1.67; 95%CI: 1.30–2.16, *p* < 0.001) in non-fixed group.

In terms of ROSC at handover at hospital, though it was not statistically significant, higher ROSC rate was achieved in fixed group in pre-period (7.5% vs. 4.3%, *p* = 0.12), and in non-fixed group in the post-period (7.8% vs. 9.4%, *p* = 0.58). The intervention significantly improved the prehospital ROSC rate in the non-fixed group (AOR: 2.37; 95%CI: 1.38–4.23, *p* = 0.002), while showed no obvious effect in the fixed group (AOR: 1.09; 95%CI: 0.50–2.42, *p* = 0.83). 

## 4. Discussion

In this metropolitan citywide intervention study, the implementation of a quality improvement programme, including a standardization of ambulance treatment protocol, targeted training of ambulance crew, and quality monitoring and feedback, improved the prehospital outcome of OHCA patients. Specifically, the intervention was associated with higher prehospital ROSC rate, and more prehospital ALS interventions such as IV access and advanced airway management. Subgroup analysis indicated that, compared with non-fixed crews, the fixed ambulance crew performed better in prehospital ALS interventions. The quality improvement programme increased the proportion of prehospital IV access, advanced airway management, and ROSC in non-fixed ambulance crew, while a non-significant improvement was seen in the fixed group.

The implementation of continuous quality improvement is an alternative approach to improving OHCA survival, which has been advocated and proved by many EMS systems [7,8,14]. Our findings added to the existing evidence that guideline adherence, coordination of tasks, team-focused resuscitation, and feedback are all instrumental in helping the EMS crews with improved treatment quality [9,10,11]. The adoption of standardized protocol is important as it provides a highly organized team treatment guide for chaotic resuscitation effort in most underperforming crews. Comparing with the proportions of pre-hospital drug administration and defibrillation in countries of high ROSC rates, such as Australia (40.4%/60.9%) [14], Germany (54.0%/25.1%) [23], and Japan (22.3%/59.2%) [24], the proportions of epinephrine administration (0.8% vs. 1.8%) and defibrillation (5.6% vs. 4.9%) remain comparatively low in China. As a result, continuous emphasizing, and training on implementing reasonable resuscitation interventions should be improved.

Another vital contributor of our interventions is that the protocol requires ambulance crews to provide on-scene resuscitation until either ROSC or termination of efforts. Despite the requirement, the on-scene time in Suzhou was only about 5 min, which was 20–40 min shorter than that reported in high ROSC countries such as North America [25], Korea [17], Germany [26] or Australia [27]. Previous studies suggested to stabilize patients on-scene before transport to hospital to avoid the fraction of time without intervention during transport [10,28]. In this sense, the marked difference of on-scene interval would explain why the ALS interventions were less used and prehospital ROSC was less often achieved in Suzhou. This lack of implementation of protocol was led by several issues: first, in most cases the patient’s family showed strong demand for transfer to hospital due to an inherent public perception that fast-transfer is the best; second, ambulance crews held the idea of fast-transfer under the pressure of insufficient ambulance resources; third, there was no clear requirement or limit of the on-scene interval during quality monitoring and feedback. It was needed to further optimize the protocol, emphasize quality monitoring, and conduct targeted community education for public.

It has been underlined by studies that the timing of ALS was more important than the provision of ALS interventions [25,29,30]. In this study, though we found increased proportion of ALS interventions as well as prehospital ROSC after the quality improvement programme, the results of subgroups showed that a higher proportion of ALS did not necessarily mean a higher proportion of ROSC. The more decisive factor in Suzhou may be the timing of treatment: the prehospital interval of about 28 min was comparable to other published data, however the response time of more than 10 min was much longer than that of North America (5–6 min) [25], Australia (7–8 min) [14], Korea (7–8 min) [17], or Japan (6–9 min) [31]. More critically, though the interval from symptom onset to call was shortened than what we reported in 2015 (30 min) [22], it was still 15–20 min. Considering the probability of survival following cardiac arrest drops 10% per minute with no intervention [32], this 15–20 min delay means that most patients lost their chance of survival during waiting, and any effort to improve on-scene ALS measures will be of no help. Our findings, along with others [8,33,34], indicate that each link in the chain of survival should be strengthened, and that efforts to involve every individual in the whole community to save patients are always warranted.

The size, pattern, level of specialty and training of ambulance crews are considered to influence the outcomes of patients with OHCA [15,16,17]. In Suzhou, the physician in a fixed crew is a dedicated prehospital personnel who only works on ambulance, while the physician in a non-fixed unit could be rotated from in-hospital emergency department or other in-hospital departments. The proportions of prehospital ALS interventions were higher in the fixed ambulance crews, which might be due to by the higher level of team coordination, and the more experience of physicians to deal with OHCA cases. In a recent systematic review, researchers emphasized that higher exposure to attempted resuscitation cases, but not years of clinical EMS experience, was associated with improved patient outcome [18]. Although the baseline difference did exist, the performance of non-fixed crews could be largely improved after use of standardized protocol and team-based training, which increased their “exposure” to simulated OHCA treatment. We suggested that, instead of finding a perfect “fixedness” of ambulance crews, the protocol adherence and targeted training are needed.

This paper reported the process and result of Suzhou’s first citywide quality improvement programme. The effectiveness of our results should be interpreted in the context of some limitations. First, because multiple interventions were simultaneously adopted, it is impossible to identify the components responsible for the observed improvement of outcomes. Second, this study used a before-and-after analytic design, which may have been affected by temporal trends in OHCA survival. Third, this was a retrospective study, which increased the risk of residual confounding. Although we eliminated imbalance using propensity score-matching analysis, unmeasured confounding factors may have influenced the outcomes. Fourth, due to limited variables in the database, the in-hospital outcomes, the quality, and time interval for each prehospital ALS interventions could not be obtained. Fifth, it was recorded in our database that all the cases had received DA-CPR and ambulance CPR, but whether any individual received bystander CPR is unknown. Last, the impact of the COVID-19 pandemic has significantly changed systems-of-care for OHCA in Suzhou, though we simply excluded cases occurring in two epidemic periods, we were unable to rule it out completely.

## 5. Conclusions

The implementation of a citywide quality improvement programme with a standardization of ambulance treatment protocol, targeted training of ambulance crew, and quality monitoring and feedback, was associated with higher prehospital ROSC rates and improved prehospital ALS interventions by ambulance crews. Though the fixed ambulance crews performed better than non-fixed crews in critical care provision and prehospital outcome, protocol adherence and targeted training could fill the underperformance of non-fixed crews efficaciously. Further improvements are needed to strengthen each link in the survival chain with a focus on ambulance treatment protocol and education.

## Figures and Tables

**Figure 1 jcm-12-00163-f001:**
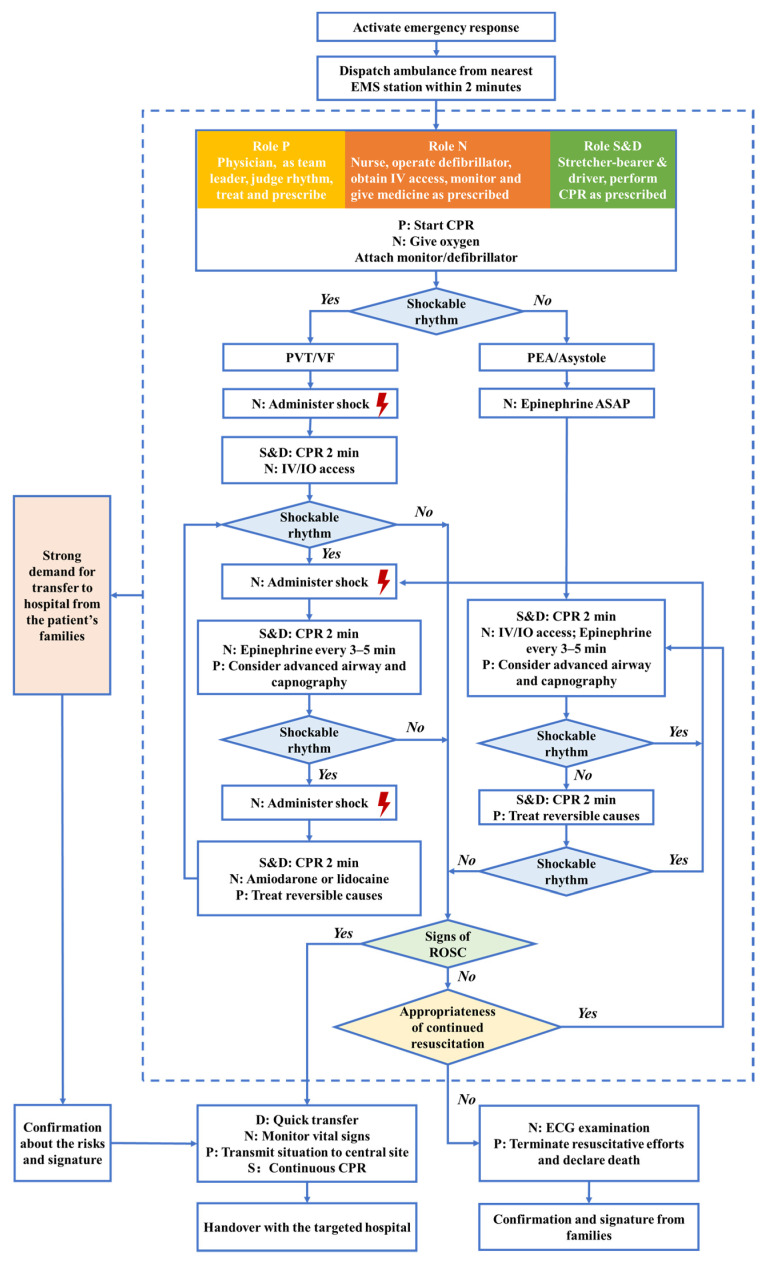
The standardized ambulance treatment protocol for OHCA patients. EMS, emergency medical services; OHCA, out-of-hospital cardiac arrest; CPR, cardiopulmonary resuscitation; PVT, pulseless ventricular tachycardia; VF, ventricular fibrillation; PEA, pulseless electrical activity; IV, intravenous; IO, intraosseous; ROSC, return of spontaneous circulation; ECG, electrocardiograph.

**Figure 2 jcm-12-00163-f002:**
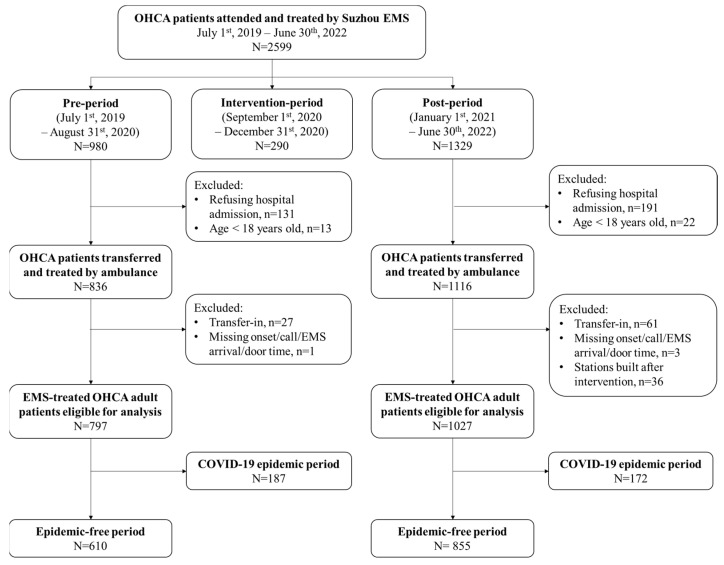
Patient flowchart. OHCA, out-of-hospital cardiac arrest; EMS, emergency medical services. There were 2 COVID-19 epidemic periods in Suzhou: 1 February 2020 to 30 April 2020 of pre-period, and 1 February 2022 to 30 April 2022 of post-period, respectively.

**Figure 3 jcm-12-00163-f003:**
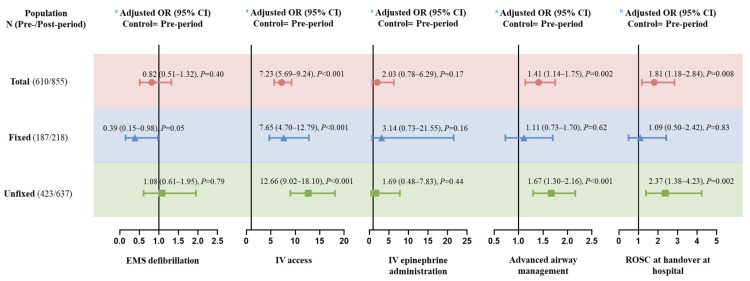
Adjusted logistic regression analysis of the effect of the improvement intervention on outcomes in EMS-treated OHCA patients by the type of ambulance crew. (ORs were calculated for post-period vs. pre-period; ^a^ Adjusted for gender, age, location, onset on weekend, call time of the day, CVD history, etiology, onset-to-call time, call-to-arrive time; ^b^ Adjusted for gender, age, location, onset on weekend, call time of the day, CVD history, etiology, onset-to-call time, call-to-arrive time, on-scene time, scene-to-hospital time.). EMS, emergency medical services; OHCA, out-of-hospital cardiac arrest; CVD, cardiovascular disease; IV, intravenous; ROSC, return of spontaneous circulation; OR, odds ratio; CI, confidence interval.

**Table 1 jcm-12-00163-t001:** Demographics, prognostic factors, and outcomes of OHCA patients during the pre-period and post-period in total cases and propensity score matched cases.

	Total Cases (N = 1465)			Propensity Score-Matched Cases ^a^ (N = 1206)			
	Pre-Period (N = 610)	Post-Period (N = 855)	*p*	Pre-Period (N = 591)	Post-Period (N = 591)	*p*	SMD
Gender = female, n (%)	154 (25.2)	221 (25.8)	0.81	150 (25.4)	148 (25.0)	0.95	0.008
Age, mean (SD)	60.88 (17.85)	62.31 (16.65)	0.12	60.83 (17.88)	60.72 (16.84)	0.91	0.006
Location = home, n (%)	372 (61.0)	530 (62.0)	0.70	360 (60.9)	358 (60.6)	0.95	0.007
Onset on weekend, n (%)	190 (31.1)	243 (28.4)	0.27	183 (31.0)	178 (30.1)	0.80	0.018
Call time of day, n (%)			0.91			0.95	0.034
0:00–5:59	84 (13.8)	114 (13.3)		82 (13.9)	78 (13.2)		
6:00–11:59	198 (32.5)	266 (31.1)		189 (32.0)	191 (32.3)		
12:00–17:59	176 (28.9)	260 (30.4)		174 (29.4)	181 (30.6)		
18:00–23:59	152 (24.9)	215 (25.1)		146 (24.7)	141 (23.9)		
CVD history, n (%)	190 (31.1)	302 (35.3)	0.10	188 (31.8)	181 (30.6)	0.71	0.026
Presumed cardiac etiology, n (%)	444 (72.8)	655 (76.6)	0.10	435 (73.6)	432 (73.1)	0.89	0.011
Onset-to-call time, min, Median (IQR)	15.0 (10.0, 20.0)	15.0 (10.0, 20.0)	0.35	15.0 (10.0, 20.0)	15.00 (10.0, 20.0)	0.26	0.046
Response interval, min, Median (IQR)	11.8 (9.1, 15.5)	11.4 (8.6, 15.2)	0.07	11.8 (9.0, 15.5)	12.0 (8.8, 15.9)	0.88	0.004
Scene interval, min, Median (IQR)	5.2 (2.6, 8.5)	5.6 (3.0, 9.2)	0.033	5.2 (2.7, 8.5)	5.1 (2.8, 8.3)	0.84	0.038
Transport interval, min, Median (IQR)	7.1 (4.4, 10.3)	8.8 (5.3, 14.2)	<0.001	7.0 (4.4, 10.3)	7.6 (4.6, 11.9)	0.23	0.069
Prehospital interval, min, Median (IQR)	25.7 (20.8, 33.4)	28.7 (22.0, 36.2)	<0.001	25.7 (20.8, 33.3)	27.4 (20.9, 34.7)	0.13	0.031
Ambulance crew type, n (%)			0.033			1	0.004
Fixed	187 (30.7)	218 (25.5)		186 (31.5)	185 (31.3)		
Non-fixed	423 (69.3)	637 (74.5)		405 (68.5)	406 (68.7)		
Treatment-EMS defibrillation, n (%)	34 (5.6)	42 (4.9)	0.63	34 (5.8)	18 (3.0)	0.032	/
Treatment-IV access, n (%)	139 (22.8)	580 (67.8)	<0.001	138 (23.4)	403 (68.2)	<0.001	/
Treatment-IV epinephrine use, n (%)	5 (0.8)	15 (1.8)	0.17	5 (0.8)	10 (1.7)	0.29	/
Treatment-airway, n (%)	296 (48.5)	492 (57.5)	0.001	291 (49.2)	337 (57.0)	0.009	/
ROSC at handover at hospital, n (%)	32 (5.2)	77 (9.0)	0.008	32 (5.4)	53 (9.0)	0.024	/

^a^ Propensity score matched for: gender, age, location, onset on weekend, call time of the day, CVD history, etiology, onset-to-call time, call-to-arrive time, on-scene time, scene-to-hospital time, call-to-hospital time, ambulance crew type. OHCA, out–of-hospital cardiac arrest; CVD, cardiovascular disease; EMS, emergency medical services; ROSC, return of spontaneous circulation; IV, intravenous; SD, standard deviation; IQR, interquartile range; SMD, standardized mean differences.

**Table 2 jcm-12-00163-t002:** Demographics, prognostic factors, and outcomes of OHCA patients treated by fixed or non-fixed EMS stations during the pre-period and post-period.

	Pre-Period (N = 610)			Post-Period (N = 855)		
	Fixed Crew (N = 187)	Non-Fixed Crew (N = 423)	*p*	Fixed Crew (N = 218)	Non-Fixed Crew (N = 637)	*p*
Gender = female, n (%)	47 (25.1)	107 (25.3)	1.0	53 (24.3)	168 (26.4)	0.59
Age, mean (SD)	61.19 (18.04)	60.74 (17.79)	0.78	64.18 (16.13)	61.67 (16.79)	0.06
Location = home, n (%)	114 (61.0)	258 (61.0)	1.0	136 (62.4)	394 (61.9)	0.94
Onset on weekend, n (%)	63 (33.7)	127 (30.0)	0.39	67 (30.7)	176 (27.6)	0.39
Call time of day, n (%)			0.45			0.66
0:00–5:59	23 (12.3)	61 (14.4)		26 (11.9)	88 (13.8)	
6:00–11:59	69 (36.9)	129 (30.5)		71 (32.6)	195 (30.6)	
12:00–17:59	49 (26.2)	127 (30.0)		71 (32.6)	189 (29.7)	
18:00–23:59	46 (24.6)	106 (25.1)		50 (22.9)	165 (25.9)	
CVD history, n (%)	57 (30.5)	133 (31.4)	0.85	79 (36.2)	223 (35.0)	0.74
Presumed cardiac etiology, n (%)	144 (77.0)	300 (70.9)	0.14	179 (82.1)	476 (74.7)	0.026
Onset-to-call time, min, Median (IQR)	20.00 (10.00, 20.00)	15.00 (10.00, 20.00)	0.018	20.00 (10.00, 20.00)	15.00 (10.00, 20.00)	0.034
Response interval, min, Median (IQR)	11.03 (8.63, 13.96)	12.45 (9.50, 16.21)	<0.001	11.80 (9.07, 15.21)	11.28 (8.47, 15.18)	0.49
Scene interval, min, Median (IQR)	5.27 (2.69, 7.74)	5.08 (2.48, 8.79)	0.93	4.94 (2.78, 7.82)	5.82 (3.03, 9.60)	0.024
Transport interval, min, Median (IQR)	7.60 (5.03, 10.38)	6.73 (4.28, 10.25)	0.049	9.47 (6.13, 13.37)	8.58 (4.98, 14.72)	0.27
Prehospital interval, min, Median (IQR)	24.80 (21.15, 32.27)	26.18 (20.74, 34.51)	0.25	28.55 (21.97, 35.14)	28.76 (22.06, 36.96)	0.33
Treatment-EMS defibrillation, n (%)	14 (7.5)	20 (4.7)	0.13	8 (3.7)	34 (5.3)	0.37
Treatment-IV access, n (%)	90 (48.1)	49 (11.6)	<0.001	189 (86.7)	391 (61.4)	<0.001
Treatment-epinephrine use, n (%)	2 (1.1)	3 (0.7)	0.65	7 (3.2)	8 (1.3)	0.07
Treatment-airway, n (%)	122 (65.2)	174 (41.1)	<0.001	149 (68.3)	343 (53.8)	<0.001
ROSC at handover at hospital, n (%)	14 (7.5)	18 (4.3)	0.12	17 (7.8)	60 (9.4)	0.58

OHCA, out-of-hospital cardiac arrest; CVD, cardiovascular disease; EMS, emergency medical services; ROSC, return of spontaneous circulation; IV, intravenous; SD, standard deviation; IQR, interquartile range.

## Data Availability

Data are available from the corresponding author on reasonable request.

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
