# Peer review of "Effect of a Targeted Ambulance Treatment Quality Improvement Programme on Outcomes from Out-of-Hospital Cardiac Arrest: A Metropolitan Citywide Intervention Study"

_jcm, 2022, doi:10.3390/jcm12010163_

Round 1
Reviewer 1 Report
Thank you for the opportunity to review the paper "Effect of a targeted ambulance treatment quality improvement program on outcomes from out-of-hospital cardiac arrest: A metropolitan citywide intervention study".
The work is worthy, but requires a major revision.
In order to place the Suzhou ambulance service and results in international comparison, the authors should add and calculate the following variables, present them in the results, and evaluate them in the discussion:
1. incidence of OHCA per 100000 population per year, overall and in the study periods.
2. incidence of OHCA with cardiopulmonary resuscitation by the ambulance service per 100000 inhabitants and year, overall and in the study periods
3. number of patients with shockable rhythm
4. number of patients with asystole/PEA
5. number of patients with witnessed OHCA
6. number of patients with OHCA witnessed by EMS /paramedic
7. CPR performed by bystander or by EMS during witnessed event
8. number of patients transported under CPR
The following outcome parameters should be added, if possible:
- Discharge rate or 30-day survival rate.
- rate of patients with "CPC1 or CPC 2" score at discharge
It is striking that the on-scene time in these Chinese study was relevantly 20 to 40 minutes shorter than, for example, in the PARAMEDIC Trials [8, 9] or in North American [3, 5, 11] or Australian[1, 12] or German resuscitation services [2, 4, 6, 7, 10]). This explains why the advanced resuscitation measures (i.v. access, advanced airway, defibrillation or epinephrine, amiodarone) were used significantly less frequently and why a ROSC was achieved significantly less often. In the above-mentioned rescue services, a patient under resuscitation is only very rarely transported to the hospital, which, on the other hand, is probably the current standard in China - Suzhou rescue service. In the above-mentioned rescue services nearly, all patients with VF will be defibrillated on scene, nearly all patients with non-shockable rhythm will receive epinephrine on scene, this does not seem to be the case in Suzhou rescue service.
In this respect, the authors should point out these differences and explain why the strategy of early transport without ALS-measures on scene is regularly followed in China and why it is assumed that this is advantageous for the patient.
I have added more reference to my review which all showed that advanced live support is different in these EMS services compared to the Suzhou rescue service.
1. Beck B, Bray JE, Smith K et al. (2016) Description of the ambulance services participating in the Aus-ROC Australian and New Zealand out-of-hospital cardiac arrest Epistry. Emerg Med Australas 28:673-683
2. Behrens NH, Fischer M, Krieger T et al. (2020) Effect of airway management strategies during resuscitation from out-of-hospital cardiac arrest on clinical outcome: A registry-based analysis. Resuscitation 152:157-164
3. Benger JR, Kirby K, Black S et al. (2018) Effect of a Strategy of a Supraglottic Airway Device vs Tracheal Intubation During Out-of-Hospital Cardiac Arrest on Functional Outcome: The AIRWAYS-2 Randomized Clinical Trial. JAMA 320:779-791
4. Burger A, Wnent J, Bohn A et al. (2018) The Effect of Ambulance Response Time on Survival Following Out-of-Hospital Cardiac Arrest. Dtsch Arztebl Int 115:541-548
5. Cheskes S, Verbeek PR, Drennan IR et al. (2022) Defibrillation Strategies for Refractory Ventricular Fibrillation. N Engl J Med
6. Gässler H, Kurka L, Rauch S et al. (2022) Mechanical chest compression devices under special circumstances. Resuscitation 179:183-188
7. Knapp J, Huber M, Gräsner J-T et al. (2022) Outcome differences between PARAMEDIC2 and the German Resuscitation Registry: a secondary analysis of a randomized controlled trial compared with registry data. European journal of emergency medicine : official journal of the European Society for Emergency Medicine
8. Perkins GD, Ji C, Deakin CD et al. (2018) A Randomized Trial of Epinephrine in Out-of-Hospital Cardiac Arrest. N Engl J Med 379:711-721
9. Perkins GD, Lall R, Quinn T et al. (2015) Mechanical versus manual chest compression for out-of-hospital cardiac arrest (PARAMEDIC): a pragmatic, cluster randomised controlled trial. Lancet 385:947-955
10. Seewald S, Wnent J, Gräsner J-T et al. (2022) Survival after traumatic cardiac arrest is possible-a comparison of German patient-registries. In: BMC emergency medicine. p 158
11. Wang HE, Schmicker RH, Daya MR et al. (2018) Effect of a Strategy of Initial Laryngeal Tube Insertion vs Endotracheal Intubation on 72-Hour Survival in Adults With Out-of-Hospital Cardiac Arrest: A Randomized Clinical Trial. JAMA 320:769-778
12. Woodall J, Mccarthy M, Johnston T et al. (2007) Impact of advanced cardiac life support-skilled paramedics on survival from out-of-hospital cardiac arrest in a statewide emergency medical service. Emerg Med J 24:134-138
Author Response
We appreciate Reviewer 1 for his/her careful review and thoughtful comments. We addressed all the comments raised by the reviewer as summarized below. The page and line numbers appearing here refer to those in the revised manuscript with tracked changes.
Comment 1: In order to place the Suzhou ambulance service and results in international comparison, the authors should add and calculate the following variables, present them in the results, and evaluate them in the discussion:
- incidence of OHCA per 100000 population per year, overall and in the study periods.
- incidence of OHCA with cardiopulmonary resuscitation by the ambulance service per 100000 inhabitants and year, overall and in the study periods
- number of patients with shockable rhythm
- number of patients with asystole/PEA
- number of patients with witnessed OHCA
- number of patients with OHCA witnessed by EMS /paramedic
- CPR performed by bystander or by EMS during witnessed event
- number of patients transported under CPR
The following outcome parameters should be added, if possible:
- Discharge rate or 30-day survival rate.
- - rate of patients with "CPC1 or CPC 2" score at discharge
Authors’ response: Thanks for the insightful comment. The reviewer has raised an important issue regarding the OHCA registry, which was not only valuable in comparing the ambulance service internationally, but also important in continuously quality improvement of the EMS itself. Suzhou EMS is building its own OHCA registry, however, during the study period of our study (July 1st, 2019 to June 30th, 2022), there was no formal registry of OHCA cases.
The data used in this study was from the emergency dispatch system and pre-hospital electronic medical record system of Suzhou EMS. The emergency dispatch system recording is generated automatically along with the dispatching process and recorded the patient's age, gender, location, chief complaint, dispatcher suspected disease, and the duration of each phase of the dispatch instructions. The pre-hospital electronic medical record is filled by ambulance physicians at the field using a tablet computer, and completed after transporting the patients to in-hospital emergency department, which contains patient’s symptoms, field diagnosis, disease history, treatment measures, pre-hospital outcome, time intervals, EMS station and ambulance personnel. Due to limitation of the data, some variables listed by the reviewer were not able to obtained.
- For variable 1 and 2: From the available data, we calculated the EMS attended and treated OHCA incidence, which was 11.8,13.0, 13.2, 13.2 per 100 000 population per year from 2019 to 2022. We have added these results in the Results section (see Page 8, Line 234-235).
- For variable 3 and 4: There was no records about patients’ rhythm before defibrillation during the study period. Suzhou EMS have noticed this lack of important variables, and is working on adding this information when optimizing the OHCA registry.
- For variable 5 and 6: All the OHCA cases included in this study were witnessed, called for EMS, and attended and treated by EMS, and patients all presented no vital signs when ambulance arrive scene. We have stated this in the Limitation section (see Page 14, Line 400-402), and have added this information in the Results section (see Page 9, Line 239-240). Suzhou EMS have noticed this lack of witness variables, and is working on adding this information when optimizing the OHCA registry.
- For variable 7 and 8: According to Suzhou EMS’s dispatch protocol, for suspected cardiac arrests, dispatcher-assisted CPR is delivered, but there were no records about whether bystander perform CPR as instructed or by him/herself. In this study, all the OHCA patients underwent CPR by EMS on-scene and during transport, so we didn’t compare these variables between pre-intervention and post-intervention period. We have stated this in the Limitation section (see Page 14, Line 402-404), and have added this information in the Results section (see Page 9, Line 239-240).
- For the 2 outcome parameters: At present, the EMS medical record and in-hospital medical record were operated by different data systems, the in-hospital outcomes could not be obtained by EMS at present. We have stated this in the Limitation section (see Page 14, Line 400-402). Suzhou EMS have noticed this lack of important variables, and is working on adding this information when optimizing the OHCA registry.
Comment 2: It is striking that the on-scene time in these Chinese study was relevantly 20 to 40 minutes shorter than, for example, in the PARAMEDIC Trials [8, 9] or in North American [3, 5, 11] or Australian[1, 12] or German resuscitation services [2, 4, 6, 7, 10]). This explains why the advanced resuscitation measures (i.v. access, advanced airway, defibrillation or epinephrine, amiodarone) were used significantly less frequently and why a ROSC was achieved significantly less often. In the above-mentioned rescue services, a patient under resuscitation is only very rarely transported to the hospital, which, on the other hand, is probably the current standard in China - Suzhou rescue service. In the above-mentioned rescue services nearly, all patients with VF will be defibrillated on scene, nearly all patients with non-shockable rhythm will receive epinephrine on scene, this does not seem to be the case in Suzhou rescue service.
In this respect, the authors should point out these differences and explain why the strategy of early transport without ALS-measures on scene is regularly followed in China and why it is assumed that this is advantageous for the patient.
I have added more reference to my review which all showed that advanced live support is different in these EMS services compared to the Suzhou rescue service.
- Beck B, Bray JE, Smith K et al. (2016) Description of the ambulance services participating in the Aus-ROC Australian and New Zealand out-of-hospital cardiac arrest Epistry. Emerg Med Australas 28:673-683
- Behrens NH, Fischer M, Krieger T et al. (2020) Effect of airway management strategies during resuscitation from out-of-hospital cardiac arrest on clinical outcome: A registry-based analysis. Resuscitation 152:157-164
- Benger JR, Kirby K, Black S et al. (2018) Effect of a Strategy of a Supraglottic Airway Device vs Tracheal Intubation During Out-of-Hospital Cardiac Arrest on Functional Outcome: The AIRWAYS-2 Randomized Clinical Trial. JAMA 320:779-791
- Burger A, Wnent J, Bohn A et al. (2018) The Effect of Ambulance Response Time on Survival Following Out-of-Hospital Cardiac Arrest. Dtsch Arztebl Int 115:541-548
- Cheskes S, Verbeek PR, Drennan IR et al. (2022) Defibrillation Strategies for Refractory Ventricular Fibrillation. N Engl J Med
- Gässler H, Kurka L, Rauch S et al. (2022) Mechanical chest compression devices under special circumstances. Resuscitation 179:183-188
- Knapp J, Huber M, Gräsner J-T et al. (2022) Outcome differences between PARAMEDIC2 and the German Resuscitation Registry: a secondary analysis of a randomized controlled trial compared with registry data. European journal of emergency medicine : official journal of the European Society for Emergency Medicine
- Perkins GD, Ji C, Deakin CD et al. (2018) A Randomized Trial of Epinephrine in Out-of-Hospital Cardiac Arrest. N Engl J Med 379:711-721
- Perkins GD, Lall R, Quinn T et al. (2015) Mechanical versus manual chest compression for out-of-hospital cardiac arrest (PARAMEDIC): a pragmatic, cluster randomised controlled trial. Lancet 385:947-955
- Seewald S, Wnent J, Gräsner J-T et al. (2022) Survival after traumatic cardiac arrest is possible-a comparison of German patient-registries. In: BMC emergency medicine. p 158
- Wang HE, Schmicker RH, Daya MR et al. (2018) Effect of a Strategy of Initial Laryngeal Tube Insertion vs Endotracheal Intubation on 72-Hour Survival in Adults With Out-of-Hospital Cardiac Arrest: A Randomized Clinical Trial. JAMA 320:769-778
- Woodall J, Mccarthy M, Johnston T et al. (2007) Impact of advanced cardiac life support-skilled paramedics on survival from out-of-hospital cardiac arrest in a statewide emergency medical service. Emerg Med J 24:134-138
Authors’ response: Thanks for the thoughtful comments and valuable suggestions. The timing of ALS and the provision of ALS interventions are both important contributors to the pre-hospital treatment of OHCA patients, which should be both emphasized.
In our study, the on-scene time was remarkably shorter than that of countries with high ROSC rates. It is true that “early transport without ALS-measures on scene” is regularly followed in Suzhou, however it was not assumed as advantageous for patient. On the contrary, it was suggested by the standardized ambulance treatment protocol of Suzhou EMS that all appropriate resuscitation efforts be made and that efforts should continue until 1) the patient be transported if good prognostic signs were present after field efforts; 2) the declaration of death; or 3) strong demand for transfer from the patient’s families. We failed to achieve enough on-scene time mainly due to three reasons:
- First, in most cases the patient’s family showed strong demand for transfer to hospital due to an inherent public perception that fast-transfer is the best;
- Second, ambulance crews held the idea of fast-transfer under the pressure of insufficient ambulance re-sources;
- Third, there was no clear requirement or limit of the on-scene interval during quality monitoring and feedback.
Among the above reasons, the strong demand for transfer to hospital by families was the main reason why “early transport without ALS-measures on scene” regularly followed in Suzhou.
Based on these findings, Suzhou EMS is working on optimizing the protocol and quality monitoring, and will conduct targeted community education for public as its continuously quality improvement intervention for the next step.
We appreciate Reviewer #1 for his/her suggestive comments, which help us modified the logic of discussion. We’ve added discussion and thought about on-scene time in our Discussion section (see Page 12, Line 314-328; and Page 13, Line 345-359; and References 25-30).
Reviewer 2 Report
Major issue:
- please specify the resuscitation protocol in Figure 1. Were there only 2-3 cycles of CPR (2 min) as shown in the figure or was there a loop?
Minor issues:
- please write the dates down in full words
- please be consistent with the outcomes in the abstract: "ROSC at handover" vs "ROSC"
- please change "timeliness" to "timelines" in row 48
- recommend changing "The EMS in China has adopted a supplementary physician or physician-paramedic model for ambulance advanced care: The prehospital medical crew of the urban EMS typically consists of one physician, one nurse, one stretcher-bearer and one driver [20,21], the crew configuration and number of personnel on ambulance is always the same." to: "The EMS in China has adopted a supplementary physician or physician-paramedic model for ambulance advanced care. The prehospital medical crew of the urban EMS typically consists of one physician, one nurse, one stretcher-bearer, and one driver [20,21]. The crew configuration and the number of personnel on the ambulance are always the same."
- might be better to define the COVID-19 epidemic period in "2.1. Study design"
- please specify all the abbreviations in Figure/Table legends
- please specify all the abbreviations at the first use and be generally consistent in using and writing words
- please specify for the readers in the "2.7. Statistical analysis" when did you use mean and when median and define P25 and P75 in words.
- please erase the instructor text in lines 254-256.
- consider changing "always" to "still" and "falls" to "drops" in row 327
- consider deleting "meaningless" in row 329
- please explain "DA-CPR" in row 342
- in row 354 please change "on strengthen" to "to strengthen" and "capacity building" to " improvement" in row 355.
- Reference 12 should be corrected
- it would be very interesting to see the neurological outcome of the survivors. Could you perhaps provide the data?
Author Response
We appreciate Reviewer 2 for his/her careful review and thoughtful comments. We addressed all the comments raised by the reviewer as summarized below. The page and line numbers appearing here refer to those in the revised manuscript with tracked changes.
Major issue: Please specify the resuscitation protocol in Figure 1. Were there only 2-3 cycles of CPR (2 min) as shown in the figure or was there a loop?
Authors’ response: There was a loop of cycles of CPR in the resuscitation protocol developed by Suzhou EMS, we have modified the Figure 1 to make the loop shown (see Page 5, Figure 1).
Minor issues:
1- please write the dates down in full words.
Authors’ response: Thanks for the kind reminder. We have modified all the dates through the manuscript and figures.
2- please be consistent with the outcomes in the abstract: "ROSC at handover" vs "ROSC".
Authors’ response: We have changed the “ROSC” to “ROSC at handover” in the abstract (see Page 1, Line 28). Also, we have added a clear definition of “prehospital ROSC” as “ROSC at handover” in the Methods section (see Page 8, Line 200).
3- please change "timeliness" to "timelines" in row 48
Authors’ response: We have changed the spiling of word “timelines” (see Page 1, Line 43; and Page 2, Line 49).
4- recommend changing "The EMS in China has adopted a supplementary physician or physician-paramedic model for ambulance advanced care: The prehospital medical crew of the urban EMS typically consists of one physician, one nurse, one stretcher-bearer and one driver [20,21], the crew configuration and number of personnel on ambulance is always the same." to: "The EMS in China has adopted a supplementary physician or physician-paramedic model for ambulance advanced care. The prehospital medical crew of the urban EMS typically consists of one physician, one nurse, one stretcher-bearer, and one driver [20,21]. The crew configuration and the number of personnel on the ambulance are always the same."
Authors’ response: The sentence has been modified as suggested (see Page 2, Line 62-63).
5- might be better to define the COVID-19 epidemic period in "2.1. Study design"
Authors’ response: Thanks for the suggestion. There were 2 Covid-19 epidemic periods in Suzhou: February 1st, 2020 to April 30th, 2020 and February 1st, 2022 to April 30th, 2022, respectively. We have added this information in Study design section (see Page 2, Line 91-91), and the legends of Figure 2 (see Page 7, Line 182-183).
6- please specify all the abbreviations in Figure/Table legends.
Authors’ response: We have added all the abbreviations in Figure/Table legends.
7- please specify all the abbreviations at the first use and be generally consistent in using and writing words.
Authors’ response: Thanks for the comments. We have checked all abbreviations at first use and make sure they were consistent in the manuscript.
8- please specify for the readers in the "2.7. Statistical analysis" when did you use mean and when median and define P25 and P75 in words.
Authors’ response: Thanks for the comments. We have added specific statistical methods for statistical description as following (see Page 8, Line 204-210).:
“Normal distribution was confirmed using the Kolmogorov–Smirnov test. Continuous variables with normal distribution were shown as mean (standard deviation, SD), and compared by student t test. Continuous variables with non-normal distribution were shown as median (interquartile range, IQR), and compared using Wilcoxon test. Categorical variables were presented as the number (percentage), and compared using chi square test.”
9- please erase the instructor text in lines 254-256.
Authors’ response: Thanks for the reminder. We have modified this.
10- consider changing "always" to "still" and "falls" to "drops" in row 327.
Authors’ response: We have changed the words as suggested.
11- consider deleting "meaningless" in row 329.
Authors’ response: We have changed the sentence as suggested.
12- please explain "DA-CPR" in row 342.
Authors’ response: We have added the full word of DA-CPR as dispatcher-assisted cardiopulmonary resuscitation at its first use (see Page 2, Line 71).
13- in row 354 please change "on strengthen" to "to strengthen" and "capacity building" to " improvement" in row 355.
Authors’ response: We have changed the words accordingly.
14- Reference 12 should be corrected.
Authors’ response: We have modified Reference 12.
15- it would be very interesting to see the neurological outcome of the survivors. Could you perhaps provide the data?
Authors’ response: Thanks for the comment. At present, the EMS medical record and in-hospital medical record were operated by different data systems, the in-hospital outcomes could not be obtained by EMS at present. We have stated this in the Limitation section (see Page 14, Line 400-402). Suzhou EMS have noticed this lack of important variables, and is working on adding this information when optimizing the OHCA registry.
Round 2
Reviewer 2 Report
Minor revisions:
- Figure 1: Please explain/correct: For example - according to the protocol in a patient with a shockable rhythm on presentation and a non-shockable rhythm and no ROSC after 2 min of CPR resuscitation should be terminated. Or another example - in a patient with a non-shockable rhythm on presentation and no ROSC after 4 min of CPR resuscitation should also be terminated. Is this correct or should the protocol figure be changed? Probably the figure should be changed so that there is a loop also for non-shockable rhythms. No arrows need to point to "Signs of ROSC" except if there is a defined number of CPR cycles before CPR termination (in this case that number should be provided).
Also in the figure legend please also provide description of abbreviations: PVT/VF and PEA.
- line 315: please consider removing "remained".
- line 317: please consider removing "relevantly".
- line 318: please correct "German" to "Germany".
- line 321: please correct "was" to "were".
- line 322: please change "ROSC was less achieved" to " ROSC was less often achieved".
- Conclusion seems too long now. I recommend: "The implementation of a citywide quality improvement programme with a standardization of ambulance treatment protocol, targeted training of ambulance crew, and quality monitoring and feedback, was associated with higher prehospital ROSC rates and improved prehospital ACLS interventions by ambulance crews. Though the fixed ambulance crews performed better than non-fixed crews in critical care provision and prehospital outcome, protocol adherence and targeted training could fill the underperformance of non-fixed crews efficaciously. Further improvements are needed to strengthen each link in the survival chain with a focus on ambulance treatment protocol and education."
Author Response
We appreciate Reviewer #2 for his/her valuable comments. We addressed all the comments raised by the reviewer as summarized below.
Minor revisions:
- Figure 1: Please explain/correct: For example - according to the protocol in a patient with a shockable rhythm on presentation and a non-shockable rhythm and no ROSC after 2 min of CPR resuscitation should be terminated. Or another example - in a patient with a non-shockable rhythm on presentation and no ROSC after 4 min of CPR resuscitation should also be terminated. Is this correct or should the protocol figure be changed? Probably the figure should be changed so that there is a loop also for non-shockable rhythms. No arrows need to point to "Signs of ROSC" except if there is a defined number of CPR cycles before CPR termination (in this case that number should be provided).
Also in the figure legend please also provide description of abbreviations: PVT/VF and PEA.
Authors’ response: Thanks for the comments. We have noticed the missing loop for non-shockable rhythms and modified the protocol. For the arrows to “Signs of ROSC”, though we didn’t define number of CPR cycles before CPR termination, we asked the ambulance crew to consider appropriateness of continued resuscitation if there were no signs of ROSC. Please see the new Figure 1, and all the abbreviations have been provided.
Suzhou EMS has noticed the need for a more practical protocol, and is working on the TOR protocol, we believe that the valuable comments from Reviewer 2 will help us a lot.
Comments:
- line 315: please consider removing "remained".
- line 317: please consider removing "relevantly".
- line 318: please correct "German" to "Germany".
- line 321: please correct "was" to "were".
- line 322: please change "ROSC was less achieved" to " ROSC was less often achieved".
Authors’ response: Thanks for the reminder. We have modified all the above wording mistakes.
Comments:- Conclusion seems too long now. I recommend: "The implementation of a citywide quality improvement programme with a standardization of ambulance treatment protocol, targeted training of ambulance crew, and quality monitoring and feedback, was associated with higher prehospital ROSC rates and improved prehospital ACLS interventions by ambulance crews. Though the fixed ambulance crews performed better than non-fixed crews in critical care provision and prehospital outcome, protocol adherence and targeted training could fill the underperformance of non-fixed crews efficaciously. Further improvements are needed to strengthen each link in the survival chain with a focus on ambulance treatment protocol and education."
Authors’ response: Thanks. We have shortened the conclusion part following the Reviewer’s recommendation.